# Targeting Oxidative Stress with Polyphenols to Fight Liver Diseases

**DOI:** 10.3390/antiox12061212

**Published:** 2023-06-03

**Authors:** Ivo F. Machado, Raul G. Miranda, Daniel J. Dorta, Anabela P. Rolo, Carlos M. Palmeira

**Affiliations:** 1CNC—Center for Neuroscience and Cell Biology, CIBB—Center for Innovative Biomedicine and Biotechnology, University of Coimbra, 3000 Coimbra, Portugal; imachado@cnc.uc.pt (I.F.M.); anpiro@ci.uc.pt (A.P.R.); 2IIIUC—Institute of Interdisciplinary Research, University of Coimbra, 3000 Coimbra, Portugal; 3School of Pharmaceutical Science of Ribeirão Preto, University of São Paulo, São Paulo 14040, Brazil; raul.miranda@usp.br; 4Department of Chemistry, Faculty of Philosophy, Sciences and Letters of Ribeirão Preto, University of São Paulo, Ribeirão Preto 14040, Brazil; djdorta@ffclrp.usp.br; 5Department of Life Sciences, University of Coimbra, 3000 Coimbra, Portugal

**Keywords:** liver, oxidative stress, ROS, mitochondria, polyphenols, quercetin, resveratrol, curcumin

## Abstract

Reactive oxygen species (ROS) are important second messengers in many metabolic processes and signaling pathways. Disruption of the balance between ROS generation and antioxidant defenses results in the overproduction of ROS and subsequent oxidative damage to biomolecules and cellular components that disturb cellular function. Oxidative stress contributes to the initiation and progression of many liver pathologies such as ischemia-reperfusion injury (LIRI), non-alcoholic fatty liver disease (NAFLD), and hepatocellular carcinoma (HCC). Therefore, controlling ROS production is an attractive therapeutic strategy in relation to their treatment. In recent years, increasing evidence has supported the therapeutic effects of polyphenols on liver injury via the regulation of ROS levels. In the current review, we summarize the effects of polyphenols, such as quercetin, resveratrol, and curcumin, on oxidative damage during conditions that induce liver injury, such as LIRI, NAFLD, and HCC.

## 1. Introduction

Oxidative stress develops when the production of reactive oxygen species (ROS) overwhelms the cellular antioxidant scavenging system. Although they are indispensable signaling molecules, excessive concentrations of ROS disturb cellular signaling [1,2]. ROS is a term used to describe a group of molecules derived from the incomplete reduction of molecular oxygen (O_2_). The free radicals superoxide (O_2_^•−^) and hydroxyl radical (^•^OH) and the nonradical hydrogen peroxide (H_2_O_2_) are the most common ROS. Superoxide and hydrogen peroxide mainly originate from the mitochondrial electron transport chain [3,4] and from enzymes including NADPH oxidases and cytochrome P450 (CYP) [5,6]. In addition, ROS production can be triggered by exposure to environmental stressors including nutrients, drugs, and ionizing radiation such as ultraviolet (UV) and X-ray [7].

In recent years, the physiological role of oxidants has been a subject of much debate. The detection, measurement, and discrimination of ROS pose a major challenge [8] and ROS have antagonistic physiological roles because they are both beneficial and harmful. For example, moderate levels of mitochondrial ROS (insufficient to cause damage) improve health and longevity by inducing adaptive responses through a mechanism called mitohormesis [9]. On the contrary, oxidative damage caused by exaggerated levels of ROS impairs the function of biomolecules due to peroxidation of lipids and oxidation of proteins and DNA, ultimately resulting in irreversible cell death [10]. Oxidative stress impairs cellular and tissue functions by causing inflammation [11,12], programmed cell death [13,14,15,16], and mitochondrial malfunction [17,18]. To protect against oxidative damage, organisms have developed defenses aimed at regulating ROS levels and preserving redox homeostasis throughout their evolution. These defense mechanisms rely upon the capability of antioxidant enzymes to detoxify oxidants and repair oxidative damage. For example, superoxide dismutase (SOD) converts superoxide into oxygen and hydrogen peroxide, catalase (CAT), and peroxidases, which convert hydrogen peroxide into water and glutathione (GSH) peroxidase (GPx), which convert oxidants into more stable compounds.

Oxidative stress is implicated as a primary cause of disease and a contributor to disease progression [19]. The non-physiological production of ROS leads to the exhaustion of antioxidant defenses and is associated with many pathologies, including Alzheimer’s disease [20], cancer [21], type 2 diabetes mellitus (T2DM) [22], aging [23], and liver diseases [24]. The loss of hepatic function that occurs during many liver conditions, such as hepatotoxicity [25], liver ischemia-reperfusion (I/R) injury (LIRI) [19], non-alcoholic fatty liver disease (NAFLD) [26], and hepatocellular carcinoma (HCC) [27], is associated with oxidative stress due to excessive production of ROS. Liver dysfunction involves the disruption of whole-body homeostasis; it causes disturbances in inter-organ crosstalk and can lead to death [28]. Recovering liver function or ameliorating the severity of liver pathologies is crucial to improving the health and life expectancy of patients with such diseases. Oxidative stress is a common factor in liver pathologies; therefore, understanding the molecular mechanisms underlying redox signaling in liver diseases will contribute to the development of novel therapeutic strategies to prevent them and ameliorate their complications. The investigation of natural products, such as polyphenols, is an attractive and promising strategy for modulating oxidative stress in liver diseases. Polyphenols, found in plants and fruits, have been shown to have a protective role against environmental stressors and to regulate ROS-mediated signaling pathways [29,30]. In the current review, we describe the effects of quercetin, resveratrol, and curcumin on the amelioration of oxidative damage that occurs during NAFLD, LIRI, and HCC and discuss how they can be attractive therapeutic compounds under such conditions.

## 2. Cellular Mechanisms of Redox Signaling

Initially, ROS generation was generally viewed only as a byproduct of aerobic processes, and exposure to these oxidants was considered detrimental to organisms. This view has changed in recent years as ROS have been found to be versatile pleiotropic signaling molecules that are required to guarantee cellular homeostasis. However, the intracellular concentration of ROS is crucial to determine their effect on cells. At low to moderate levels, ROS act as second messengers and participate in a plethora of biological processes, such as cellular proliferation and differentiation [31,32], cell cycle progression [33], immune system function [34], metabolism [35], the formation of oxygenated polyunsaturated fatty acids (oxylipins) [36], and the regulation of adaptive responses to environmental stress cues [9,37]. On the other hand, excessive levels of ROS disrupt redox signaling, severely damage cellular components, and promote cell death. Therefore, ROS levels must be tightly regulated to avoid tissue injury. Furthermore, because of their complex role in physiological signaling, a deep understanding of the cellular mechanisms underlying redox signaling is essential for the development of therapeutic strategies involving the modulation of ROS levels.

Redox signaling typically involves the reversible reduction and oxidation of proteins, interfering with their activity, stability, and cellular localization. Oxidants can modulate the function of kinases, phosphatases, receptors, and transcription factors (Figure 1). For instance, the specificity of hydrogen peroxide against sulfur atoms on the side chains of cysteine and methionine residues allows it to regulate protein function [38,39]. Signals triggered by hydrogen peroxide can therefore be amplified due to the activation of protein cascades that eventually lead to the alteration of gene expression [40,41]. For example, protein tyrosine phosphatases (PTPs) are inactivated by hydrogen peroxide owing to the oxidation of thiol groups (-SH) in the side chains of cysteine residues to sulfenic acid (-SOH) [42,43]. Upon the formation of sulfenic acid, disulfide bonds (-S-S-) can be formed in the catalytic center of PTPs [44], which disrupts their interaction with their protein substrates, resulting in tyrosine phosphorylation activity [45]. However, further oxidation of sulfenic acid to sulfinic acid (-SO_2_H) and sulfonic acid (-SO_3_H) is irreversible [46]. Hydrogen peroxide is involved in the regulation of the insulin signaling cascade by reversibly inactivating the protein tyrosine phosphatase 1B (PTP1B) [47,48]. In turn, PTP1B-mediated dephosphorylation of the insulin receptor is impaired [47,49]. Furthermore, adaptive cell survival pathways are activated by hydrogen peroxide through the activation of the phosphatidylinositol 3-kinase (PI3K)/AKT pathway [50]. Hydrogen peroxide has been found to inactivate phosphatase and tensin homolog (PTEN), thus inducing the downstream signaling of PI3K [51,52].

Redox homeostasis is also preserved by biological systems that sense oxidants and use transcriptional regulation mechanisms to modulate their levels. Under physiological conditions, nuclear factor erythroid 2-related factor 2 (Nrf2) is maintained at a low abundance in the cell owing to repression by Kelch-like ECH-associated protein 1 (KEAP1) [53]. When oxidant levels rise, KEAP1 undergoes conformational changes due to the oxidation of its cysteine residues, preventing the targeting of Nrf2 for proteasomal degradation [54]. Nrf2 is translocated to the nucleus, where it activates the transcription of many cytoprotective and antioxidant target genes [54]. The transcription factor nuclear factor-κB (NF-κB) is a regulator of many cellular processes including cell proliferation, inflammation, apoptosis, and stress responses [55]. Depending on the context, NF-κB can either be activated or repressed by oxidants that influence its stress response. Hydrogen peroxide leads to the activation of inhibitor of NF-κB (IκB) kinases and the consequent phosphorylation of IκB, resulting in the activation of NF-κB [56,57,58], whereas the oxidation of cysteine residues on NF-κB caused by increased nuclear hydrogen peroxide generation represses it [59].

Proteins involved in energy and nutrient sensing, such as adenosine monophosphate (AMP)-activated protein kinase (AMPK), are also subject to regulation by redox signaling. AMPK is activated in response to alterations in the intracellular AMP-to-adenosine trisphosphate (ATP) ratio that result from decreased ATP generation due to nutrient deprivation, hypoxia, or deficient mitochondrial oxidative generation [60]. Upon its activation, AMPK modulates metabolism with the aim of preserving energy under ATP-limiting conditions. With this purpose, it phosphorylates downstream targets to increase catabolism and decrease anabolism. Glucose metabolism, lipid metabolism, autophagy, and mitochondrial homeostasis are some of the major pathways modulated by AMPK [60]. Initially, it was proposed that AMPK was activated directly by ROS-mediated oxidation [61,62]; however, a recent study found that AMPK activity is modulated indirectly by ROS due to ROS-mediated changes to mitochondrial ATP generation [63].

Excessive levels of oxidants lead to oxidative damage to cellular components, biomolecules, and DNA, interfering with redox signaling and disrupting cellular function. Cells possess many mechanisms to protect themselves from oxidative stress, including antioxidant enzymes, which are the front-line of defense against ROS [64]. For instance, SODs are metalloenzymes that catalyze the reduction of superoxide to less toxic hydrogen peroxide and molecular oxygen. In mammals, copper/zinc-SOD (CuZn-SO, SOD1) is found in the cytosol [65], whereas manganese-SOD (Mn-SOD, SOD2) and extracellular SODs (ecSOD, SOD3) are found in the mitochondrial and extracellular matrices, respectively [66,67]. The compartmentalization of different SOD isoforms allows the local regulation of redox signaling [68]. The hydrogen peroxide resulting from the dismutation of superoxide can be further decomposed into molecular oxygen and water by CAT, which can manage hydrogen peroxide levels and redox signaling [69]. In addition, GPx is a class of peroxidases involved in the degradation of hydrogen peroxide to water and in the degradation of lipid peroxides to their corresponding alcohols. The reaction catalyzed by GPx involves the oxidation of GSH to glutathione disulfide (GSSG), which is subsequently reduced back to GSH by glutathione reductase. For example, GPx plays an important role in protecting against lipid peroxidation by activating stimulator-of-interferon genes (STING) [70].

## 3. Intracellular Sources of ROS

### 3.1. Mitochondria

Mitochondria are crucial players in guaranteeing whole-body homeostasis as they are responsible for many functions that drive most biological pathways and are vital for normal cellular and organismal functions. They provide energy as ATP through oxidative phosphorylation and regulate glucose and lipid metabolism, calcium signaling, autophagy, apoptosis, and ROS signaling [71]. These tasks are accomplished because there are established communication routes between the mitochondria and other organelles, such as the nucleus. Mitonuclear communication allows the activation of nuclear gene expression by the mitochondria in response to mitochondrial stress. Since most mitochondrial genes are encoded in the nuclear genome, efficient mitonuclear communication is essential for proper cellular and mitochondrial functioning. The mitochondrial DNA (mtDNA) contains a small fraction of mitochondrial genes that encode subunits of mitochondrial oxidative phosphorylation, transfer RNAs, and subunits of the mitochondrial ribosome. The remaining mitochondrial proteins are encoded by nuclear DNA and imported from the cytosol into the mitochondria. Mitochondrial well-being is also maintained by mitochondrial quality control mechanisms, including mitochondrial bioenergetics, dynamics (fusion and fission events), and mitophagy. In fact, cells rely on the tight coordination between these mechanisms to survive and adapt to different environmental stresses. Mitochondrial biogenesis, fusion/fission, and mitophagy all contribute to the renovation of the cell’s mitochondrial pool. The synthesis of new mitochondria through mitochondrial biogenesis requires the activation of the master regulator peroxisome proliferator-activated receptor gamma coactivator 1-alpha (PGC-1α), which activates nuclear respiratory factors (NRF1 and NRF2) and mitochondrial transcription factor A (TFAM), which is responsible for the transcription and replication of mtDNA. The newly formed mitochondria are fused to form the mitochondrial network through fusion events that require the participation of mitofusins 1 and 2 (MFN1 and MFN2), and optic atrophy protein 1 (OPA1). On the other hand, the clearance of defective mitochondria through mitophagy requires the involvement of dynamin-related protein 1 (DRP1) and mitochondrial fission 1 protein (FIS1). There are two main regulatory pathways of mitophagy, which are ubiquitin-mediated mitophagy, which involves tensin homolog-induced putative kinase 1 (PINK1) and Parkin, and receptor-mediated mitophagy. These mechanisms are important for the organization and shape of the mitochondrial network within the cell, which influences energy generation and mitonuclear communication [72]. In fact, mitochondrial morphology is thought to be linked to mitochondrial adaptation during stress [73]. The dynamics of the mitochondrial network influence ROS production [74]. A balance between mitochondrial dynamics and bioenergetics must prevail for mitochondrial homeostasis. Under stress, such as in the context of disease, mitochondrial quality control mechanisms are overwhelmed, leading to the accumulation of defective mitochondria and a deficiency in energy production. Indeed, mitochondrial dysfunction is associated with many pathologies, including NAFLD, LIRI, neurodegenerative diseases, cancer, and even age-related diseases.

The energy that fuels biological processes is primarily synthesized through the oxidative phosphorylation that occurs within mitochondria. The reducing equivalents, nicotinamide adenine dinucleotide (NADH) and Flavin adenine dinucleotide (FADH_2_), generated by glycolysis and the tricarboxylic acid (TCA) cycle fuel the electron transport chain. NADH-ubiquinone oxidoreductase (complex I) and succinate dehydrogenase (complex II) transfer electrons from NADH and FADH_2_, respectively, to ubiquinone, which is subsequently reduced to ubiquinol. Electrons from ubiquinol are transferred to cytochrome *c* via the cytochrome *bc*_1_ complex (complex III). Finally, cytochrome *c* oxidase (complex IV) transfers electrons to molecular oxygen to generate water. The increasing affinity of these complexes for electrons is coupled with the energetically unfavorable pumping of protons from the mitochondrial matrix to the intermembrane space, leading to the generation of an electrochemical gradient across the mitochondrial inner membrane that stores potential energy [75]. F_1_F_O_ ATP synthase takes advantage of this stored energy to phosphorylate adenosine diphosphate (ADP), thereby generating ATP while transferring protons back to the mitochondrial matrix. Under both physiological and pathological conditions, electrons prematurely leak from electron transport, mainly from complex I and complex III, and react with oxygen, leading to the formation of superoxide and hydrogen peroxide. In total, there are approximately 11 sites of ROS generation in mitochondria [76]. When oxidants reach supraphysiological levels, oxidative damage to the cellular and mitochondrial components occurs. For instance, complexes of the electron transport chain are particularly susceptible to damage by oxidants, leading to the disruption of oxidative phosphorylation. Oxidative damage leads to mtDNA lesions, such as thymine glycols and 8-oxoguanine [77,78]. Cardiolipin, a phospholipid present in mitochondrial membranes that regulates bioenergetics, dynamics, and morphology [79], is subjected to peroxidation by oxidants [80]. Oxidants trigger the opening of the mitochondrial permeability transition (MPT) pore, which is associated with increased apoptosis [81].

In contrast, low to mild levels of ROS are involved in mitochondrial signaling pathways, including mitochondrial quality control mechanisms. AMPK, a regulator of mitochondrial biogenesis and mitophagy, is indirectly activated by ROS [63]. It has been shown that PGC-1α stimulates the transcription of antioxidant proteins to protect mitochondria against oxidative damage [82]. Mitochondrial morphology and function are also regulated by ROS, which modulate proteins such as OPA1, DRP1, and MFN2, which are required for the fission and fusion of the mitochondria [83]. Mild levels of superoxide were shown to trigger DRP1- and Parkin/PINK1-dependent mitophagy, without triggering non-selective mitophagy [84,85]. This suggests that ROS are involved in the removal of dysfunctional mitochondria. In addition, it has been suggested that superoxide protects mitochondria against oxidative stress by leading to a decrease in mitochondrial ROS through a mechanism that involves an interaction between superoxide and uncoupling proteins (UCPs) [86]. UCPs are mitochondrial inner membrane proteins that pump protons from the mitochondrial matrix to the mitochondrial intermembrane space, dissipating the proton gradient as heat [87]. Transient mitochondrial stress involving low to moderate levels of oxidants has also been shown to trigger adaptive mitochondrial responses [9,88,89]. Caloric restriction and exercise are known inducers of ROS and have been found to improve health and lifespan by triggering mitohormetic responses [90,91]. Increased ROS levels lead to the activation of mitochondrial antioxidant defenses, which include antioxidant enzymes, such as CAT and SOD [92]. Mitohormetic stimuli have also been shown to induce mitochondrial biogenesis, dynamics, and mitophagy [9].

### 3.2. Cytochrome P450

The liver plays an important role in the detoxication of drugs and other xenobiotics. The biotransformation of xenobiotics requires many enzymes such as CYPs, hydroxylases, reductases, and peroxidases. CYPs are monooxygenases that comprise most of the phase I reaction enzymes involved in the metabolism of xenobiotics and are predominantly located in the liver [93]. They also participate in other cellular processes such as the biosynthesis of bile acids, metabolism of eicosanoids, and biosynthesis of steroid hormones [94]. The binding of a substrate to the ferrous catalytic center of CYPs sets the beginning of the reaction. Subsequently, CYP accepts an electron from NADPH reductase and binds to an oxygen molecule [95]. The reduction of this complex to a peroxy complex is then followed by protonation, which results in the cleavage of an O–O bond and the formation of water and a highly reactive complex [95]. The uncoupling of electron transfer from NADPH to CYP with monooxygenation of the substrate is associated with increased production of ROS [96]. Although CYPs are necessary to metabolize xenobiotics and prevent their harmful effects, they can also exacerbate organismal damage by inducing oxidative stress. Interestingly, oxidative stress has also been shown to regulate the transcription of CYPs. For instance, the expression of *CYP1A1* is repressed by hydrogen peroxide [97]. Furthermore, increased generation of ROS due to the induction of CYP2E1 leads to the enhancement of cellular antioxidant defenses, such as increased GSH and CAT levels [98].

## 4. Targeting Oxidative Stress with Natural Compounds in Liver Pathophysiology

Since ancient times, the use of products from natural sources in therapy has been reported [99,100]. Indigenous populations and ancestral civilizations have always used natural products, usually from plants or animals, to prevent or treat disease [101]. Archaeological findings dating back over 5000 years report the use of medicinal plants in therapy. This is also reported in other historical documents, such as the Ebers Papyrus of the Egyptians, Chinese Medical Material of the Chinese, and Ayuved of the Indians [102]. The use of natural products in therapy has been resurfacing and gaining space in modern medicine, not only for the treatment and prevention of diseases but also as a complementary therapy to the conventional therapies widely used today. In addition, studying natural products can bring a scientific view to something that has always been used by ancestors as “popular knowledge” to understand and evaluate their health benefits.

Natural products play an important role in the discovery of new drugs by generating a rich library of chemical structures with a wide range of potential biological effects. The great advantage of natural products is that most have associated biological activities [103].

With the development of organic and synthetic chemistry, there is a growing search for isolating new substances from natural sources, such as plants, animals, and even microorganisms [100], as well as synthetic routes for structural modifications of previously isolated molecules that are known to improve chemical characteristics, increase the therapeutic effect, and decrease the toxic potential of the base molecule [104]. However, the ease of access to some phytochemicals (natural products from plants) and their use in the form of extracts and as nutraceuticals (food and/or food components with potential medicinal use) without laborious pharmaceutical syntheses have attracted the attention of scientists for the therapeutic use of natural products [103]. Many biological activities associated with natural products have already been reported, such as antibiotic [102], antiparasitic [105], antifungal [106], antioxidant [107], anti-inflammatory [108], antitumorigenic [109], and hepatoprotective activities [110]. Thus, it is an important direct and indirect source for the development of new drugs.

Natural products have long been used for the prevention and/or treatment of liver diseases in several countries. Several phytochemicals have been identified and have shown therapeutic potential in various liver conditions [111]. These compounds have specific characteristics of action with intrinsic benefits and/or low toxicity [103]. Natural compounds with antioxidant [112], anti-inflammatory [113], and anticarcinogenic properties [109] have been shown to be hepatoprotective with few adverse effects. Thus, they can serve as a primary source for the development of new drugs or as complementary therapies for liver diseases [100,103,114]. In the following sections, we discuss one of the main classes of natural products with potential use in liver conditions—polyphenols.

### 4.1. Oxidative Stress in Liver Pathophysiology

#### 4.1.1. Liver Ischemia/Reperfusion Injury

The many functions of the liver include the removal of toxic substances from the blood, regulation of glucose metabolism, and participation in immune responses. Disruption of hepatic function has adverse effects on organisms and may result in liver failure [115,116]. Liver damage during liver resection or transplantation often implies an ischemic insult that results from an interruption in blood flow. The period of anoxia is also associated with nutrient deprivation resulting in increased ROS levels and decreased ATP generation. Once the blood flow is restored, the hepatic tissue is reoxygenated. However, paradoxically, damage that may have started during ischemia is exacerbated by reperfusion. During this phase, mitochondrial ROS from hepatocytes reach exaggerated levels, leading to the oxidative damage of DNA, proteins, and lipids [117]. Oxidative phosphorylation then becomes inefficient and starts to generate even more ROS. In addition, these events trigger the opening of the MPT pore, leading to apoptosis and the necrosis of hepatocytes. During reperfusion, injured liver cells release ROS and damage-associated molecular patterns (DAMPs) [118], which trigger the activation of a proinflammatory immune response. Upon activation, Kupffer cells recruit monocytes and neutrophils to further induce the proinflammatory immune cascade. These immune cells release proinflammatory cytokines such as tumor necrosis factor α (TNFα), interleukin-1β (IL-1β), IL-6, and IL-8. The inflammatory response during reperfusion is self-sustained by the continuous production of cytokines and ROS, which further exacerbate LIRI [118,119]. In addition, during I/R, ROS production favors the release of the nuclear DAMP high mobility group box-1 protein (HMGB1) [120], which acts as a mediator of inflammation and contributes to liver damage in a mechanism involving the activation of NF-κB signaling and toll-like receptor 4 (TLR4) [121]. Furthermore, the ablation of TLR9 protected mice from LIRI in part by limiting the production of cytokines and ROS in neutrophils [122].

#### 4.1.2. Non-Alcoholic Fatty Liver Disease

NAFLD is composed of two types of liver diseases: non-alcoholic fatty liver (NAFL) and non-alcoholic steatohepatitis (NASH). Although not the most common, patients with NAFL can develop NASH, which increases the risk of developing cirrhosis and HCC, conditions that usually require liver transplantation for their treatment [123]. Unhealthy lifestyle habits, such as sedentarism and the overconsumption of nutrients, are important risk factors for the development of NAFLD [124]. The accumulation of at least 5% of triglycerides in hepatocytes (hepatic steatosis) characterizes NAFLD and leads to inflammation and hepatocyte death [125]. Excessive body fat results in the death of adipocytes, which secrete cytokines and inflammatory factors with the concomitant release of free fatty acids in hepatocytes. These events lead to the development of insulin resistance [125,126]. Consequently, aberrant triglyceride synthesis in hepatocytes ensues, leading to mitochondrial uncoupling, excessive generation of ROS, and the activation of cell death receptors, which increase the liver’s susceptibility to injury [125,126]. Increased ROS generation contributes to NAFLD progression by inducing oxidative damage to the mitochondrial electron transport chain, therefore resulting in lower ATP production [127] by oxidizing cardiolipin in the inner mitochondrial membrane [128], leading to the activation of proapoptotic and proinflammatory pathways [129]. Furthermore, the antioxidant capacity of patients with NASH is suggested to be decreased since they have low serum levels of SOD [130]. However, a study demonstrated that NAFLD patients instead have increased activity of antioxidant enzymes [131].

#### 4.1.3. Hepatocellular Carcinoma

HCC is the most prominent type of liver cancer, and its development is associated with hepatitis B virus (HBV), NAFLD, and cirrhosis [132]. While the mechanisms underlying HCC are still not completely understood, it is thought that the risk factors for progression into HCC involve genetic predisposition, lifestyle habits such as alcohol and tobacco consumption, HBV and hepatitis C virus (HCV) infection, and liver diseases. Oxidative stress has been shown to be related to the development of HCC due to its damaging roles concerning DNA, cell proliferation, and apoptosis, which are all important players in the development of cancer. In fact, lesions and mutations of the DNA, including mutations of the genes *TERT*, *TP53*, and *CTNNB1*, are primary drivers of the initial phase of HCC [133]. For instance, 8-hydroxy-2′-deoxyguanosine (8-OHdG) is a mutational marker associated with increased oxidative stress that also serves as a biomarker for HCC [134]. In addition, oxidative stress induces the release of proinflammatory cytokines and chemokines that further contribute to the development of HCC. The malignancy of HCC may be determined due to the balance between oxidative stress and antioxidant responses [135].

### 4.2. The Therapeutic Potential of Polyphenols against Liver Diseases

Polyphenols are a class of substances that are ubiquitous in plants and which can be considered one of the most abundant classes of substances with antioxidant potential in the diet [136,137,138]. They are present in many types of food and plant such as vegetables, cereals, teas, red wine, coffee, fruits, flowers, medicinal plants, and mushrooms [137]. More than 8000 molecules have already been identified [112,138] as secondary metabolites that present great variations in structural size, ranging from small molecules with low molecular weight to polymerized molecules with up to 50 monomer units [139]. Chemically, polyphenols have an aromatic ring, with one or more hydroxyl groups attached to that ring. According to their chemical structure, they can be divided into flavonoids, lignans, stilbenes, phenolic acids, and curcuminoids [109,137].

A growing body of evidence has shown the human benefits of dietary intake of polyphenols [114], such as the prevention and treatment of diabetes [138], obesity [140,141], metabolic and cardiovascular diseases [140,142], cancer [114], and liver disorders [137]. The advantages of the use of polyphenol compounds for human disorders are their accessibility (polyphenols are easily found in foods and plants), specificity of response, and, in most cases, lower toxicity. However, the bioavailability and rapid metabolism of polyphenols can become the main problem associated with their use. Many factors can influence the concentration of polyphenols in food and plants, including environmental and biochemical factors [112,114,143]. Among the polyphenols, we focus on flavonoids (quercetin), stilbenes (resveratrol), and curcuminoids (curcumin) (Figure 2), which show therapeutic potential for liver diseases because of their antioxidant and anti-inflammatory actions, in addition to their low toxicity. These may be alternatives for the treatment of liver disorders as well as for adjuvant treatments.

#### 4.2.1. Quercetin

The most abundant polyphenols in plants and food are flavonoids, which are a large class of secondary metabolites in plants, and there is evidence that they function as antioxidants in plants associated with environmental stress [144,145]. Because of this property and its abundance in the human diet, attention has been drawn to its antioxidant and therapeutic properties in humans [145]. Chemically, flavonoids are polycyclic compounds with a basic structure of 15 carbons (C_6_-C_3_-C_6_) with two aromatic rings (known as A and B rings) connected by three carbon bridges (C-ring). Indeed, flavonoids can be sub-classified according to the oxidation state of their C-ring, hydroxylation pattern of the ring structure, and substitution of the three positions [146]. The sub-classifications of flavonoids are flavonols, flavones, isoflavones, flavanones, anthocyanidins, and flavanols [139,147,148]. Antioxidant activity is one of the main biological activities of flavonoids and depends on their chemical structure, B-ring, and substituents. The presence of a 2,3-double bond in conjugation with a 4-oxo function, *o*-di-OH on the B-ring, and 3- and 5-OH groups on the heterocyclic ring, similar to quercetin, can impart interesting antioxidant activity to the flavonoid structure [149].

Quercetin (3,3′,4′,5,7-pentahydroxyflavone) is an important flavanol that is abundantly present in broccoli, onions, berries, apple skins, herbs, tea, red grapes, and red wines [150,151]. Several studies using in vitro and in vivo experimental models have shown the benefits of quercetin against liver steatosis and NAFLD [111,152,153] (Table 1). Quercetin can also exhibit a strong antioxidant effect, even when forming a complex or pharmaceutical formulation (e.g., glycoside-conjugated, non-encapsulated) for therapeutic purposes [150,154,155]. Other activities associated with quercetin include anti-inflammatory [113] and anti-ischemic [156] effects, the attenuation of metabolic disorders [157], and the prevention of cancer [153] (Table 1).

The molecular action of quercetin is associated with its strong antioxidant activity, which plays an important role in the prevention and treatment of many diseases, including NAFLD [150]. Evidence shows that its strong antioxidant effect is due to substituents, including the 4-oxo group conjugated with 2,3-alkene, 3- and the 5-hydroxyl group as the B-ring with the *o*-di-OH group [149]. This chemical structure provides quercetin with a dual role as an antioxidant compound at lower concentrations, while at higher concentrations, it can act as a pro-oxidant, exerting an anticancer effect [166]. Indeed, recent studies have shown that autophagy plays a pivotal role in their therapeutic properties [153,167,168,169]. 

Oxidative stress due to excessive ROS production is an important factor in the progression of NAFLD, which can cause serious damage to the antioxidant system. Furthermore, the increase in ROS levels promotes the production of TNFα, increases mitochondrial damage, and even initiates the inflammatory cascade that can aggravate the inflammatory process of NAFLD [170]. Therefore, the antioxidant and anti-inflammatory properties of quercetin play critical roles.

Vidyashankar et al., using HepG2 cells pretreated with oleic acid to induce a fatty liver condition, showed that quercetin (10 µM) was able to decrease triacylglycerol (TGA) content (45%) and increase insulin-mediated glucose uptake. Moreover, quercetin reduced the inflammatory process by inhibiting *Tnfα* gene expression and attenuated the induced oxidative process by increasing cellular antioxidants, suggesting that treatment with 10 µM of quercetin effectively reduced oleic acid-induced hepatic steatosis symptoms in HepG2 cells [158]. Another study using both in vitro and in vivo models demonstrated the therapeutic properties of quercetin in NAFLD induced by T2DM. *db*/*db* mice treated with 100 mg/kg/day of quercetin for 8 weeks showed improved dyslipidemia, relieved liver swelling, restored abnormal liver enzymes, and reduced lipid accumulation and hyperglycemia. Moreover, the authors demonstrated that quercetin treatment attenuates oxidative and inflammatory processes via activation of the farnesoid X receptor 1 (FXR1)/Takeda G protein-coupled receptor 5 (TGR5) signaling pathway in both animal and cellular models [159].

Oxidative stress, calcium homeostasis, cell swelling, proinflammatory cytokines, and apoptotic and autophagic processes play critical roles in LIRI; therefore, quercetin has emerged as an interesting drug for its treatment [108,160]. In this context, Wu and colleagues showed that quercetin was able to reduce LIRI in BALB/c mice by reducing serum liver enzymes and histopathological liver damage. Furthermore, in vitro tests using primary hepatocytes showed that quercetin inhibited the release of proinflammatory cytokines (including TNFα and IL-6), suggesting that the inhibition of TNFα mediated an anti-apoptotic and anti-autophagic effect via the extracellular signal-regulated kinase (ERK)/NF-κB, which can help in LIRI treatment [108]. Similarly, other studies have demonstrated a protective effect of quercetin against LIRI. Uylaş et al. showed that in rats, the dose is important in quercetin treatment. While a low dose (25 mg/kg) of quercetin was not effective, a higher dose (100 mg/kg) produced an adverse effect, and a mild dose (50 mg/kg) had a protective effect against LIRI [160]. The antioxidant characteristics of quercetin also play an important role in the protective effects against LIRI. Atef et al. showed that quercetin was able to reduce oxidative stress, hepatic degeneration, inflammatory cytokines, and hemeoxygenase-1 (HO-1) (which improves I/R injury) in Wistar rats, demonstrating the hepatoprotective effect of quercetin [161]. Finally, in a recent study by Ferreira-Silva et al., a liposomal nanoformulation of quercetin resulted in an in vitro and in vivo enhanced therapeutic effect on LIRI [171].

The ability of quercetin to modulate apoptotic and autophagic processes, arrest the cell cycle, and inhibit the tumor growth factor, in addition to its dual role in the oxidative process, makes quercetin an interesting drug for the prevention and treatment of HCC (conventional or in combination). Moreover, quercetin can modulate numerous transduction signals by interacting with specific proteins and receptors [166,172,173]. Several studies have investigated the antitumor potential of quercetin alone or in combination with conventional chemotherapy [174]. Yamada et al. used the HuH7 cell line, a hepatocarcinoma cell line, and found that quercetin was able to suppress the migration of HCC cells through the attenuation of AKT signaling, and that suppression of the hepatocyte growth factor (HGF) and transforming growth factor-α (TGF-α) are involved in the molecular mechanism [162]. Another in vitro study by Jeon et al. showed that the regulation of intracellular ROS levels by quercetin plays a critical role in the antiproliferative effect in HepG2. The decrease in intracellular ROS levels due to the antioxidant properties of quercetin can regulate the proliferation of HepG2 cells independently of p53 (a tumor suppressor protein) [163]. In the same way, in vivo studies also demonstrate the anti-tumor effect of quercetin. For example, Ji et al. showed that quercetin inhibited the growth of HCC via the modulation of apoptosis and autophagy in mice. Quercetin stimulated autophagy in part by inhibiting the AKT/mammalian target of the rapamycin (mTOR) pathway and activating the mitogen-activated protein kinase (MAPK) pathways (involved in cancer progression and the regulation of cellular processes) [164]. Similarly, another in vitro and in vitro study by Wu et al. using H22, HepG2, and BALB/c mice showed that quercetin was able to inhibit HCC proliferation and migration and promote apoptosis in a dose-dependent manner. After treatment by gavage, quercetin produced a reduction in the volume and weight of liver tumors in BALB/c mice through macrophage polarization in coordination with the NF-κB pathway to modulate the autophagic process [165].

#### 4.2.2. Resveratrol

Stilbenes are a large class of polyphenols that are widely distributed in food and medicinal plants. Their basic chemical structure is composed of two benzenic rings interconnected by a diene group (1,2-diphenylethene) known as (E)-stilbene. Many substances belong to the stilbene class. This class is composed of the substitution of hydrogen by a hydroxyl group and includes not only a monomeric form but also polymeric and heteromeric forms [175]. Stilbenes exhibit biological activities associated with antitumor, anti-inflammatory, antioxidant, and antibiotic effects [176,177,178].

Resveratrol (3,5,4′-trihydroxystilbene) is a polyphenol present in plants (berries), legumes (peanuts), and fruits (red grapes) [179]. It demonstrates an antineoplastic capacity by inducing apoptosis in altered cells, with a low level of toxicity to normal cells at certain doses, alongside a response observed in various tumors, including lung carcinoma, hematological malignancies (acute myelogenous leukemia and multiple myeloma), pancreatic cancer, and prostate cancer [180]. Resveratrol acts on several cellular pathways, including the modulation of cell cycle signaling factors, inhibition of the cell cycle and apoptosis, and inhibition of angiogenesis and metastasis [143,181] (Table 2).

It is believed that the combination of cytotoxic antineoplastic agents and phytochemical inhibitors may contribute to the inhibition of tumor growth by combining the mechanisms of action in HCC. Ismail et al. evaluated the effect of resveratrol and thymoquinone (TQ) both individually and in combination in HepG2 cells [186]. TQ is one of the constituents of *Nigella sativa*, with anticancer potential described for its action as an inhibitor of cell proliferation by interrupting the cell cycle, inducing apoptosis, and inhibiting angiogenesis. In vitro, both TQ and resveratrol decreased cell viability in HepG2 through the stimulation of caspase 3 activity, with the response being more intense in cells treated with resveratrol than in cells treated with thymoquinone. In cells in which the substances were combined, the effect on caspase activity increased in a dose-dependent manner, contributing to its antineoplastic potential. In this study, the generation of ROS was measured based on the intracellular GSH content after treatment with TQ and resveratrol. In cells treated with the associated substances, there was a significant reduction in GSH levels, demonstrating the release of ROS in cells. Moreover, after treatment with resveratrol and TQ, the cells showed a decrease in the amount of malondialdehyde (MDA), reducing lipid peroxidation, which may imply that the amount of ROS produced is not sufficient to affect MDA [187].

In addition to the impact and therapeutic possibilities of HCC, the effect of resveratrol has been the subject of studies on hepatic steatosis and cardiovascular indices. It has long been known that in the process of insulin resistance and dyslipidemia, which is often associated with diabetes, obesity, and NAFLD, conditions such as inflammation and oxidative stress are central to pathogenesis. Recent studies have shown that resveratrol can act beneficially in relation to blood glucose, tissue response to insulin, oxidative stress, and inflammation [182,188,189]; additionally, other studies have suggested that it has the potential capacity to reduce the severity of NAFLD, with an impact on obesity and dyslipidemia [190,191,192,193]. However, the effects of resveratrol use on cardiovascular risk factors and the potential reduction of risk-associated events remain inconclusive, and none of the studies has evaluated the impact on steatosis in patients with concomitant T2DM. Sangouni et al. carried out a double-blind, placebo-controlled, randomized clinical trial (RCT) in an in vivo model of patients with T2DM, with the objective of evaluating the effect of resveratrol supplementation (1000 mg/day) for 8 weeks on hepatic steatosis and cardiovascular indices, using the visceral fat index (VAI), lipid accumulation product (LAP), and Castelli risk index (CRI-I) as impact parameters for the aforementioned conditions and atherogenic coefficient (AC), respectively. However, the study did not demonstrate significant improvements in liver and cardiovascular indices under these conditions in patients with T2DM [194]. The results were compared with those of existing studies on resveratrol supplementation with 500 mg/day for 3 months in patients with NAFLD but without T2DM [182,195] in which a reduction in the degree of steatosis was observed, raising the hypothesis that, at least partially, the results may be related to the time of use. Regarding cardiovascular indices, even in studies in which supplementation was longer (12 weeks), no improvement was observed in cardiovascular indices in general, a parameter that uses lipid profile values for measurement [183]. This observation is consistent with the results of a more recent meta-analysis, which found that resveratrol did not affect dyslipidemia in the evaluated studies after evaluating RCTs [196]. Despite this, it is important to note that the studies in question did not consider aspects regarding the bioavailability of the substance, which makes further evaluations necessary to determine the impact of the substance on the condition.

Previous studies on LIRI and resveratrol have demonstrated a possible action of resveratrol on signaling mediated by TLR4 receptors on the surface of immune cells (monocytes, macrophages, and dendritic cells) and hepatocytes, with the consequent inhibition of NF-κB activation factors. It is known that this activation pathway is part of an important pathway for the release of proinflammatory cytokines, and it is believed that the inhibition of this signaling pathway could be key to controlling one of the molecular targets potentially associated with LIRI [197,198]. The inflammatory response induced by hepatic damage is, in part, mediated by eicosanoids and other oxylipins that act as proinflammatory factors [199]. He and colleagues investigated the potential inhibition of the TLR4/NF-κB pathway by resveratrol in hepatocytes using in vitro and in vivo models. The results obtained in the present study demonstrated that in BRL-3A cells, resveratrol at concentrations of 5, 10, or 20 μM was able to decrease cell viability associated with hypoxia/reoxygenation, in addition to reducing lactate dehydrogenase (LDH) release and apoptosis. Resveratrol inhibited the activity of TLR4 receptors, which was compatible with the inhibition of NF-κB translocation to the nucleus. Resveratrol also decreased the inflammatory cytokines produced during the pathogenesis of LIRI, especially TNFα and IL-1β [184]. In in vivo models of rats exposed to I/R resveratrol had a protective effect on liver cells by decreasing apoptosis, aspartate transaminase (AST), and alanine transaminase (ALT) levels. The results obtained in the in vivo model also demonstrated lower levels of TLR4 and NF-κB, and the serological dosages of TNFα and IL-1β were lower in rats exposed to resveratrol. However, as shown in a previous study, no strong positive correlation was obtained with the concentrations used, suggesting that only concentrations of 0.25–20 μM can protect BRL-3A cells, while concentrations higher than this safety level are capable of producing deleterious toxic effects [200].

A recent study by Wang et al. explored another pathway associated with the pathogenesis of LIRI, which involves the action of neutrophils recruited to the liver after the reperfusion process [185]. Neutrophil recruitment and infiltration produce ROS, with the consequent production of cytokines and chemokines in the inflammatory cascade that act by recruiting even more neutrophils to the parenchyma through positive feedback, intensifying liver damage [201,202]. Resveratrol was shown to improve LIRI by suppressing the neutrophil response through the modulation of the ERK signaling pathway. Oxidative stress in neutrophils was reduced and, consequently, the oxidation of lipids was reduced, and antioxidant enzymes were increased. In addition to these findings, the extracellular networks formed by neutrophils (NETs), which are part of innate immunity, were also found in smaller amounts in a LIRI mouse model [185].

#### 4.2.3. Curcumin

Curcuminoids are natural phenolic compounds (a small class of polyphenols) that include curcumin, demethoxycurcumin, and bisdemethoxycurcumin, all of which are isolated from *Curcuma longa* (turmeric) [203,204]. They are yellow in color and are commonly used as spices, pigments, and additives in food. Moreover, these compounds are of medical interest because of their potential antioxidant, anti-inflammatory, and anticarcinogenic effects [137,203]. Curcumin is a major representative compound of curcuminoids.

Curcumin (diferuloylmethane, (1E,6E)-1,7-bis(4-hydroxy-3-methoxyphenyl)-1,6-heptadiene-3,5-dione) is a yellow pigment isolated from the root and rhizome of *Curcuma longa* and *Curcuma domestica*. It is a major active curcuminoid (phenolic compound) present in turmeric that has multiple pharmacological effects [204,205].

The antioxidant and anti-inflammatory effects of curcumin play important roles in its therapeutic properties. Huang et al. investigated the antioxidant potential of curcumin formulation encapsulated in zein/casein-alginate nanoparticles under in vitro simulated gastrointestinal digestion and found that encapsulated curcumin was mainly released in the small intestine phase, increasing its bioavailability (5,7-fold higher than non-encapsulated). In hydrogen peroxide-induced oxidative stress in HepG2 cells, curcumin reduced ROS, increased SOD and CAT activities, and reduced MDA accumulation [206]. Thus, curcumin may be an interesting ally for the treatment of some liver disorders (Table 3).

In LIRI, for example, the antioxidant and anti-inflammatory effects of curcumin play a critical role in hepatoprotection. Ibrahim et al. investigated the effects of curcumin and dimethyl fumarate (DMF), independently or in combination, on LIRI. They found that both curcumin and DMF were able to protect the liver from I/R-induced injury by reducing serum liver enzyme (AST and ALT) levels, but a combination of curcumin and DMF showed a better response to these outcomes. The antioxidant and anti-inflammatory effects of curcumin, DMF, and their combination were evaluated. Curcumin improved neutrophil infiltration and the inflammatory cascade by significantly decreasing TNFα, inducible nitric oxide synthase (iNOS), IL-1β, and IL-6 levels when compared with the I/R insult group, but the combination with DMF was more effective for this purpose. Moreover, curcumin also improved the expression of Nrf2 and HO-1 in the liver following I/R injury. Thus, curcumin and its combination with DMF effectively attenuated LIRI in rats, promoting hepatoprotection through antioxidative and anti-inflammatory effects [208]. Another recent study by Zhu et al. obtained similar results in a LIRI model using C57BL/6J mice. They found that curcumin inhibits NETs formation, which can aggravate liver injury, activate an innate immune response, and alleviate LIRI by inhibiting the MAPK kinase (MEK)/ERK pathway [209].

NAFLD is another pathology in which curcumin can be an interesting candidate as a therapeutic option owing to its oxidative and inflammatory nature. An in vivo study by Li et al. using C57BL/6 mice fed a high-fat diet (HFD) and a controlled diet investigated the effect of curcumin supplementation on metabolic adaptation. They found that curcumin supplementation (0.2% curcumin for 10 weeks) significantly reduced fat mass, hepatic steatosis, and circulating lipopolysaccharide levels. Dietary curcumin improved insulin sensitivity in HFD-fed mice [207]. A systematic review and meta-analysis by Ngu et al. compiled 16 random clinical trials with a total of 1028 patients in the meta-analysis with the primary outcomes of NAFLD severity, liver steatosis resolution, liver scarring, liver enzymes, and lipid profiles after the use of curcumin as an adjuvant treatment in patients with NAFLD [211]. The findings showed that patients treated with curcumin showed an improvement in NAFLD severity and a higher prevalence of liver steatosis resolution compared with those given a placebo based on ultrasonographic findings. Moreover, curcumin was able to significantly reduce ALT, AST, and total cholesterol, although low-density lipoprotein (LDL), high-density lipoprotein (HDL), and triglycerides were not different when compared to the placebo [211].

Curcumin has potential therapeutic applications for HCC. Several studies have shown the anti-HCC effect of curcumin. Khan et al., using HCC cell lines (SMMC-7721 and Hep3B) and nude mice injected with Hep3B, demonstrated that curcumin alone and in combination with N-n-butyl haloperidol iodine (F2) derived from haloperidol and used for myocardial I/R injury inhibits malignant proliferation and migration and induces apoptosis both in vitro and in vivo in HCC [210]. The association of curcumin with other drugs (conventional drugs and phytochemicals) is an interesting alternative in the treatment of HCC. Miyazaki et al., using an in vitro model of cellular resistance to Lenvatinib and spheroids of the HCC cells line, showed that the EGFR-PI3K-AKT signaling pathway is involved in Lenvatinib resistance, and curcumin was able to suppress this signaling pathway. Thus, the co-administration of Lenvatinib and curcumin suppressed cellular proliferation, invasion, colony formation, and spheroid formation ability via suppressing the cancer stemness marker, indicating that curcumin was able to reduce Lenvatinib resistance as a therapeutical target for HCC [212]. A study by Srivastava et al. showed a synergistic effect of the association of curcumin and quercetin in multiple cancer cells. They demonstrated in vitro that curcumin in co-administration with quercetin strongly inhibited cancer cell proliferation and, in the A375 cell line, the association was able to modulate Wnt/β-catenin signaling and apoptosis by Bcl-2, caspase-3/7, and PARP pathways [213]. Similarly, the co-administration of curcumin with resveratrol [214], curcumin with piperine (a plant alkaloid from black and long pepper) [215], and curcumin with epigallocatechin-3-gallate (a major catechin in green tea) [216] showed an enhancement of the anti-cancer effect of curcumin alone. In this way, although there is an anti-proliferative/anti-cancer effect of curcumin, there are combinations with other drugs that may enhance this effect and act as a synergism against HCC.

## 5. Concluding Remarks

ROS play a dual role in organism physiology, whereas moderate levels of mitochondrial ROS can improve health and longevity by inducing adaptive responses (mitohormesis) and regulating cell signaling. However, excessive ROS levels can cause oxidative damage, impairing the function of biomolecules such as lipids (lipid peroxidation), proteins (oxidation of proteins), and DNA, which can result in irreversible cell death. Oxidative stress affects cellular and tissue functions by interfering with mitochondrial function, thereby triggering inflammation, autophagy, and programmed cell death (apoptosis). Moreover, oxidative stress has been implicated as a primary cause of disease and a stimulator of its progression. The liver is an essential organ that responds to multiple functions, including the regulation and production of metabolites and the maintenance of body homeostasis. The biotransformation of xenobiotics is easily affected by oxidative stress, and many liver disorders are associated with oxidative stress and damage. Therefore, understanding the molecular mechanisms involved in the pathways of oxidative stress and the non-physiological production of ROS in liver diseases will allow for a better understanding of the pathophysiology of the disease, help develop strategies for the management and treatment of the disease, and contribute to health extension.

Natural compounds, despite their historical use, have emerged as alternative therapeutic agents mainly because of their characteristic biological activity. They are found in plants, animals, and even microorganisms, and play an important role in drug discovery, generating a rich library of chemical structures with a range of potential biological effects. Several phytochemicals (natural compounds present in plants) have been identified and have shown therapeutic potential for liver disorders because of their antioxidant and anti-inflammatory properties, modulation of cellular processes such as autophagy and apoptosis, significant hepatoprotection, and low toxicity. Thus, natural compounds can serve as primary sources for developing new drugs or even complementary/adjuvant therapies for hepatic diseases.

Polyphenols are a large class of phytochemicals comprised of mainly flavonoids, stilbenes, and curcuminoids. The therapeutic properties of three polyphenols, quercetin (a flavonoid), resveratrol (a stilbene), and curcumin (a curcuminoid), were reviewed, with a focus on the molecular and cellular mechanisms of their potential use as therapeutics for some liver diseases. These polyphenols have great potential as therapeutic targets in oxidative stress, inflammation, and the crosstalk between them, as well as in the modulation of apoptosis and autophagy (in the case of quercetin), as evidenced by pre-clinical/experimental studies using both in vitro and in vivo models of NAFLD, HCC, and LIRI. Given the therapeutic potential of the three polyphenols reviewed, in addition to current trends in the use of natural products, the present review sheds light on some scientific evidence concerning the use of quercetin, resveratrol, and curcumin to treat liver disorders. Such compounds may be important as an alternative and complementary treatment for liver diseases.

## Figures and Tables

**Figure 1 antioxidants-12-01212-f001:**
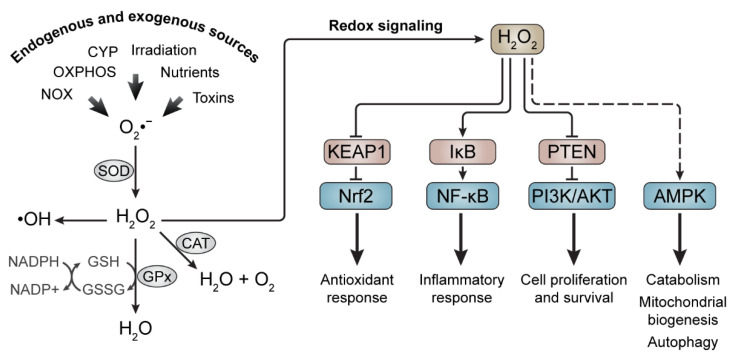
Generation of reactive oxygen species (ROS) and redox signaling. Many endogenous and exogenous sources favor the generation of superoxide and hydrogen peroxide. Enzymes that comprise the cellular antioxidant system (SOD, CAT, and GPx) transform these oxidants into less dangerous molecules. Low to moderate levels of hydrogen peroxide participate in several biological processes including the regulation of antioxidant response, inflammatory response, cell proliferation and survival, metabolic adaptation, and autophagy. Indirect processes are indicated by dashed arrows. AMPK, AMP−activated protein kinase; CAT, catalase; CYP, cytochrome P450; GPx, glutathione peroxidase; GSH, glutathione; GSSG, glutathione disulfide; IκB, inhibitor of NF−κB; KEAP1, Kelch−like ECH−associated protein 1; NF−κB, nuclear factor−κB; NOX, NADPH oxidase; Nrf2, nuclear factor erythroid 2−related factor 2; OXPHOS, oxidative phosphorylation; PI3K, phosphatidylinositol 3−kinase; PTEN, phosphatase and tensin homolog; SOD, superoxide dismutase.

**Figure 2 antioxidants-12-01212-f002:**
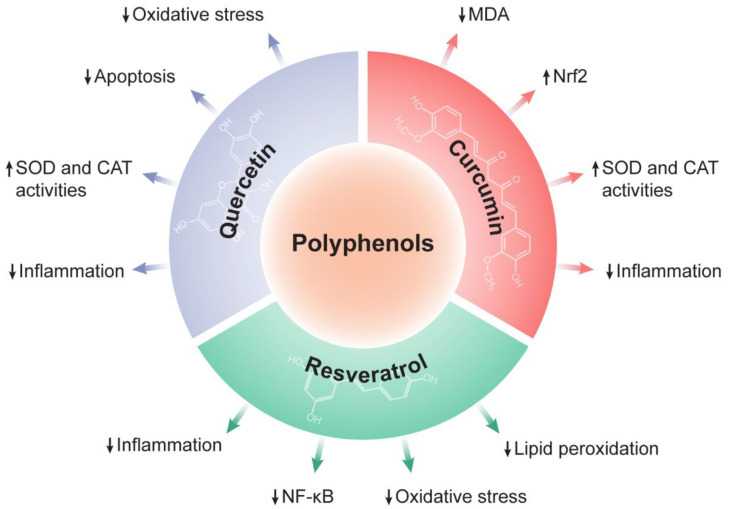
Modulation of ROS-related factors by polyphenols. In the context of liver pathology, quercetin, curcumin, and resveratrol have been found to modulate oxidative stress markers and oxidative stress-related pathways. ↑, increase; ↓, decrease; CAT, catalase; MDA, malondialdehyde; NF-κB, nuclear factor-κB; Nrf2, nuclear factor erythroid 2-related factor 2; SOD, superoxide dismutase.

**Table 1 antioxidants-12-01212-t001:** Summary of the effect of quercetin on the amelioration of liver diseases.

Model	Dosage	Mechanisms	Effects	Reference
NAFLD				
HepG2 cells	10 µM (for 24 h)	↓ TNFα ↑ Antioxidant defenses	Improved insulin-mediated glucose uptake Reduced inflammation	Vidyashankar et al. [158]
C57BLKS *db*/*db* mice	100 mg/kg/day (for 8 weeks)	↑ Antioxidant defenses ↑ FXR1/TGR5 signaling	Improved dyslipidemia Relieved liver swelling and liver enzymes Reduced lipid accumulation and hyperglycemia	Yang et al. [159]
HepG2 cells	10 and 20 µM (for 24 h)			
Liver I/R injury				
BALB/c mice	100 and 200 mg/kg/day (for 5 days)	↓ ALT and AST ↓TNFα and IL-6 ↑ p62 ↓ BECN1 and LC3 ↑ Bcl-2 ↓ Bax, CASP3 and CASP9 ↓ ERK/NF-κB pathway	Reduced serum liver enzymes Reduced histopathological liver damage Inhibited the release of proinflammatory cytokines Inhibited autophagy and alleviated apoptosis	Wu et al. [108]
Primary hepatocytes	20 µM (for 24 h before I/R)			
Spraque Dawley rats	50 mg/kg (for 30 min before I/R)	↓ AST and ALT ↓ MDA	Restored abnormal liver enzymes Caused liver histological improvement	Uylaş et al. [160]
Wistar rats	50 mg/kg (before I/R)	↑ GSH, SOD, and CAT ↓ MDA ↑ Bcl-2 ↓ TNFα, NF-κB, and HO-1	Reduced oxidative stress Reduced hepatic degeneration Reduced inflammatory cytokines	Atef et al. [161]
Hepatocellular carcinoma				
HuH7 cells	3 to 7 µM (for 1 h)	↓ AKT signaling ↓ HGF ↓ TGFα	Suppressed the migration of HCC cells	Yamada et al. [162]
HepG2, HuH7, PLC/PRF-5 and Hep3B cells	80 µM (for 24 and 48 h)	↓ ROS ↓ Cyclin A and CHK1 ↑ HO-1	Reduced proliferation of HCC cells	Jeon et al. [163]
BALB/c nude mice	60 mg/kg/day	↓ AKT and mTOR ↑ MAPK	Inhibited the growth of HCC Stimulated autophagy Induced apoptosis	Ji et al. [164]
SMMC7721, HepG2 and LO2 cells	0 to 120 µM (for 24, 36, and 48 h)			
Old BALB/c mice	25, 50, and 100 mg/kg/day (for 21 days)	↓ MMP-2 and MMP-6 ↓ NF-κB ↓ TNFα, IL-6, and IL-17A	Inhibited HCC proliferation and migration Promoted apoptosis Produced a reduction in the volume and weight of liver tumors	Wu et al. [165]
H22 and HepG2 cells	25, 50, and 100 µM (for 24, 48, and 72 h)			

↑, increase; ↓, decrease; AKT, protein kinase B; ALT, alanine transaminase; AST, aspartate transaminase; Bax, Bcl-2-Like protein 4; Bcl-2, B-cell lymphoma-2; BECN1, beclin-1; CASP3, caspase 3; CAT, catalase; CHK1, checkpoint kinase 1; ERK, extracellular signal-regulated kinase; FXR1, farnesoid X receptor 1; GSH, glutathione; HCC, hepatocellular carcinoma; HGF, hepatocyte growth factor; HO-1, hemeoxygenase-1; IL-6, interleukin-6; I/R, ischemia-reperfusion; LC3, microtubule-associated protein 1A/1B light chain 3; LIRI, liver ischemia-reperfusion injury; MDA, malondialdehyde; MMP, matrix metalloproteinase; NAFLD, non-alcoholic fatty liver disease; NF-κB, nuclear factor-κB; p62, sequestosome-1; ROS, reactive oxygen species; SOD, superoxide dismutase; TGA, triacylglycerol; TGFα, transforming growth factor alpha; TGR5, Takeda G protein-coupled receptor 5; TNFα, tumor necrosis factor α.

**Table 2 antioxidants-12-01212-t002:** Summary of the effect of resveratrol on the amelioration of liver diseases.

Model	Dosage	Mechanisms	Effects	Reference
NAFLD				
Adult patients with hepatic steatosis	500 mg/day (for 12 weeks)	↓ ALT and AST ↓ NF-κB ↓ Cytokeratin-18 ↓ Bilirubin, HDL- cholesterol, and Apo a_1_	Improved tissue response to insulin Alleviated oxidative stress and inflammation	Faghihzadeh et al. [182] Farzin et al. [183]
Liver I/R injury				
Sprague Dawley rats	10 and 20 mg/kg (for 1 h before I/R)	↓ TLR4 ↓ NF-κB ↓ p65 ↑ IκBα ↓ TNFα and IL-1β ↓ AST, ↓ ALT, and LDH	Decreased cell viability associated with hypoxia/reoxygenation Inhibited the activity of TLR4 receptors Reduced apoptosis	He et al. [184]
BRL-3A cells	0 to 100 µM (for 2 and 18 h before I/R)			
C57BL/6 mice	25 mg/kg (for 1 week, 2 h, and 30 min before I/R)	↓ NET release ↓ ROS production ↑ GSH and GPx activity ↓ ERK/c-Fos signaling	Ameliorated LIRI Inhibited the function of neutrophils Restrained neutrophils-mediated inflammatory response	Wang et al. [185]
Hepatocellular carcinoma				
HepG2 cells	64.5 µM (for 24 h)	↑ CASP3 ↓ GSH and MDA	Decreased cell viability in HepG2 Induced apoptosis	Ismail et al. [186]

↑, increase; ↓, decrease; ALT, alanine transaminase; AST, aspartate transaminase; CASP3, caspase 3; ERK, extracellular signal-regulated kinase; GSH, glutathione; GPx, glutathione peroxidase; HCC, hepatocellular carcinoma; HDL, high-density lipoprotein; IκBα, inhibitor of NF-κB α; IL-1β, interleukin-1β; I/R, ischemia-reperfusion; LDH, lactate dehydrogenase; LIRI, liver ischemia-reperfusion injury; NAFLD, non-alcoholic fatty liver disease; MDA, malondialdehyde; NET, neutrophil extracellular trap; NF-κB, nuclear factor-κB; p65, REL-associated protein; ROS, reactive oxygen species; TLR4, toll-like receptor 4.

**Table 3 antioxidants-12-01212-t003:** Summary of the effect of curcumin on the amelioration of liver diseases.

Model	Dosage	Mechanisms	Effects	Reference
NAFLD				
Old C57BL/6 mice	Curcumin supplementation on HFD 0.2%(*w*/*w*) (for 10 weeks)	Gut microbiota modulation ↓ LPS levels	Reduced body weight gain and fat deposition Ameliorated insulin resistance and improved glucose tolerance Reduced hepatic steatosis	Li et al. [207]
Liver I/R injury				
Albino rats	Pretreatment with 400 mg/kg/day (for 14 days before I/R)	↓ AST and ALT ↓ TNFα, iNOS, IL-1β and IL-6 ↑ Nrf2 and HO-1	Protected the liver from I/R-induced injury Improved neutrophil infiltration and the inflammatory cascade	Ibrahim et al. [208]
Old C57BL/6 mice	Pretreatment with 100 mg/kg (for 3 h before I/R)	↓ MEK/ERK pathway ↓ ROS production ↓ TNFα, IL-1β, and IL-6	Alleviated hepatic I/R injury Reduced the production of NET Activated innate immune response	Zhu et al. [209]
Hepatocellular carcinoma				
Nude mice	50 mg/kg/day (for 3 weeks)	↓ EZH2 ↓ Wnt/β-catenin	Inhibited cell migration and invasion Inhibited tumorigenicity Induced apoptosis	Khan et al. [210]
Hep3B and SMMC-7721 cells	15 µM (for 48 h)			

↑, increase; ↓, decrease; ALT, alanine transaminase; AST, aspartate transaminase; ERK, extracellular signal-regulated kinase; EZH2, enhancer of zeste homolog 2; HCC, hepatocellular carcinoma; HFD, high-fat diet; HO-1, hemeoxygenase-1; IL-1β, interleukin-1β; I/R, ischemia-reperfusion; LIRI, liver ischemia-reperfusion injury; iNOS, inducible nitric oxide synthase; LPS, lipopolysaccharides; MAPK, mitogen-activated protein kinase; NAFLD, non-alcoholic fatty liver disease; NET, neutrophil extracellular trap; Nrf2, nuclear factor erythroid 2-related factor 2; ROS, reactive oxygen species; SOD, superoxide dismutase; TNFα, tumor necrosis factor α.

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
