# Peer review of "Targeting Oxidative Stress with Polyphenols to Fight Liver Diseases"

_antioxidants, 2023, doi:10.3390/antiox12061212_

Round 1

Reviewer 1 Report

The manuscript is focused on the role of polyphenols in the therapy of different liver diseases. After introduction (cellular mechanisms orf redox) the authors describe the role of oxidative stress in the develment of chronic liver diseases such as MAFLD or hepatocellular carcinoma, Finally the authors describr the role of different polyphenols in the prevention of liver diseases,   Points of criticism: 1. First of all the manuscript is too long and not really comprehensively written.  190 references and not even one illustration !!!    2. Polyphenols directly or indirectly interfere with metabolism by affecting oxidative stress, inflammation, immunity and liver enteric circulation. The authors should present the beneficial activities on polyphenols on the liver by schematic illustration 3. The authors should discuss the co-administration of curcumin with other phytochemicals, including resveratrol, quercetin, epigallocatechin-3-gallate, and piperine;

4. Curcumin reverses Lenvatinib resistance in HCC, and that their combination has clinical application potential for adjunctive treatment in HCC. Could you comment on this ?? 5. Please use the proper description of fatty liver disease  MAFLD and not NAFLD  

Author Response

Reviewer 1

The manuscript is focused on the role of polyphenols in the therapy of different liver diseases. After introduction (cellular mechanisms orf redox) the authors describe the role of oxidative stress in the develment of chronic liver diseases such as MAFLD or hepatocellular carcinoma, Finally the authors describr the role of different polyphenols in the prevention of liver diseases,   Points of criticism:

We thank the reviewer for the helpful suggestions. They were crucial to improve the quality of our manuscript and hope that is now suitable for publication.

All suggested corrections are highlighted in yellow in the manuscript.

  1. First of all the manuscript is too long and not really comprehensively written. 190 references and not even one illustration !!!

An additional figure and 3 tables were added to the manuscript to summarize the content of the manuscript.

  1. Polyphenols directly or indirectly interfere with metabolism by affecting oxidative stress, inflammation, immunity and liver enteric circulation. The authors should present the beneficial activities on polyphenols on the liver by schematic illustration

All beneficial activities of polyphenols on the discussed liver diseases are now summarized in Table 1 (Quercetin), Table 2 (Resveratrol) and Table 3 (Quercetin).

  1. The authors should discuss the co-administration of curcumin with other phytochemicals, including resveratrol, quercetin, epigallocatechin-3-gallate, and piperine;

The co-administration of curcumin with other phytochemicals was added to the manuscript (page 17, line 39).

  1. Curcumin reverses Lenvatinib resistance in HCC, and that their combination has clinical application potential for adjunctive treatment in HCC. Could you comment on this ??

Indeed, curcumin reverses Lenvatinib resistance in HCC and their co-administration is an attractive therapeutic strategy for the suppression of cancer. This discussion was added to the manuscript (page 17, line 40):

“Miyazaki and colleagues, using an in vitro model of cellular resistance to Lenvatinib and spheroids of the HCC cells line, showed that the EGFR-PI3K-AKT signaling pathway is involved in Lenvatinib resistance and curcumin was able to suppress this signaling pathway. Thus, the co-administration of Lenvatinib and curcumin suppressed cellular proliferation, invasion, colony formation, and spheroid formation ability via suppressing of cancer stemness marker, indicating that curcumin was able to reduce the Lenvatinib resistance as a therapeutical target for HCC [211].”

  1. Please use the proper description of fatty liver disease MAFLD and not NAFLD

Although increasing works are adopting MAFLD instead of NAFLD to better describe fatty liver diseases, there is still great debate regarding that nomenclature (PMID: 35693413, 36179795, 34530064). Since the main subject of the current review is not on fatty liver disease and considering that a full consensus about the nomenclature has not yet been reached, we decided that is best to use the classical nomenclature NAFLD in this manuscript.

Reviewer 2 Report

General comment

Doing an original literature review on oxidative stress and the effects of flavonoids, stilbenes and curcuminoids is a great challenge given the number of papers already available. In my opinion, the points covered in this work seem well written and enjoyable to read, but, this literature review is far too synthetic in its current form for a specialist journal on the subject like "Antioxidants". It seems more suited to generalist journals. Among the points that need to be completed, we can highlight

1- The need to add a summary Figure on the different mechanisms leading to oxidative stress

2- The need for a paragraph on the mechanisms involving cytochromes P450 in the formation of ROS

3- The need for a paragraph on the mechanisms involving GSH, GSH reductase, GSH peroxidase in the defence mechanisms against ROS

4- The need for a paragraph and Figure linking oxidative stress, inflammation and cytokines

5- The need to present in a Figure the mechanisms of action of the main compounds mentioned, in relation to the previous points (one Figure per compound illustrating the particularities of their action with the structure of the leader and the other compounds of the family of similar structure whose effects are similar)

6- The need to present in Tables the effects of the main compounds mentioned, with the study model, the dose and duration of use of the antioxidant, the effects measured or monitored (one Table per compound)

Detailed comments

P1L13 : « non-alcoholic FATTY liver disease » is correct for NAFLD

P1L30: from

P2L3: GSH and GSSG and enzymes involved in their formation or their use (GSH-peroxidase) should be presented here

P2L28: among the most important examples, the formation of oxylipins seems unavoidable

P2L40: formation of R-SS-R should be detailed with the mechanism that permit to counteract this alteration

A Figure on the mechanisms of ROS formation and defence mechanisms is expected

Even if the formation of ROS by mitochondria is essential, a chapter on the formation of ROS by P450 seems unavoidable

P6L3, L22: A specific paragraph linking oxidative stress, inflammation and cytokines would be welcome

Figure 1 is really too synthetic. More details on the mechanisms for understanding the effects of polyphenols in the fight against oxidative stress would be welcome

P8L16-18, P9L6-9: the relationship between these structural features and the protective mechanism should be the subject of a Figure (with quercetin and its quinone and semiquinone forms) in place of Figure 2.

P8L13, P9L16-37: the mechanisms by which ROS cause the production of pro-inflammatory cytokines should be detailed. The role of oxylipins in these processes should be summarised.

A Table presenting the effects of quercitin in vivo and in vitro models, with doses and duration of use, is expected

P10L40: As for flavonoids, a figure explaining the mechanisms by which stilbenes have a protective role against ROS is expected

P12 L5-6: Here again, this mechanism and its relationship with the formation of oxylipins should be better detailed

A Table presenting the effects of resveratrol in vivo and in vitro models, with doses and duration of use, is expected

P13L7-13: as for the previous compounds, the mechanisms of action of curcumin should be the subject of a summary figure, the role of this compound (lipoxin inhibitor) on oxylipin formation should be better presented

Author Response

Reviewer 2

Doing an original literature review on oxidative stress and the effects of flavonoids, stilbenes and curcuminoids is a great challenge given the number of papers already available. In my opinion, the points covered in this work seem well written and enjoyable to read, but, this literature review is far too synthetic in its current form for a specialist journal on the subject like "Antioxidants". It seems more suited to generalist journals. Among the points that need to be completed, we can highlight

We appreciate the comments and helpful suggestions made by the reviewer that were important to improve our work.

All suggested corrections are highlighted in yellow in the manuscript.

1- The need to add a summary Figure on the different mechanisms leading to oxidative stress

We understand the need for the figure. On Figure 1 it can now be found the mechanism of formation of ROS as well as the main mechanisms of redox signaling.

2- The need for a paragraph on the mechanisms involving cytochromes P450 in the formation of ROS

A new section (3.2. Cytochrome P450) was added to the manuscript where it is discussed the role of Cytochrom P450 as a source of ROS (page 6, line 19).

3- The need for a paragraph on the mechanisms involving GSH, GSH reductase, GSH peroxidase in the defence mechanisms against ROS

The mechanisms involving antioxidant enzymes (GSH, GSH peroxidase, GSH reductase, SOD, Catalase) are now discussed in better detail in section 2 (page 4, line 14).

4- The need for a paragraph and Figure linking oxidative stress, inflammation and cytokines

Included in Figure 1.

5- The need to present in a Figure the mechanisms of action of the main compounds mentioned, in relation to the previous points (one Figure per compound illustrating the particularities of their action with the structure of the leader and the other compounds of the family of similar structure whose effects are similar)

We understand that the mechanisms of each compound should be better illustrated. To complement Figure 2, we added 3 tables, one for each compound, summarizing their mechanisms and effects on each liver disease.

6- The need to present in Tables the effects of the main compounds mentioned, with the study model, the dose and duration of use of the antioxidant, the effects measured or monitored (one Table per compound)

A table for each compound was added to the manuscript.

Detailed comments

P1L13 : « non-alcoholic FATTY liver disease » is correct for NAFLD

P1L30: from

Corrected.

P2L3: GSH and GSSG and enzymes involved in their formation or their use (GSH-peroxidase) should be presented here

Added to the paragraph and discussed in more detail on section 2.

P2L28: among the most important examples, the formation of oxylipins seems unavoidable

Added to the examples.

P2L40: formation of R-SS-R should be detailed with the mechanism that permit to counteract this alteration

The formation of R-SS-R is now discussed in more detail on section 2 (page 4, line 41).

A Figure on the mechanisms of ROS formation and defence mechanisms is expected

Summarized in Figure 1.

Even if the formation of ROS by mitochondria is essential, a chapter on the formation of ROS by P450 seems unavoidable

Discussion on cytochrome P450 as a source of ROS was added to the manuscript (section 3.2).

P6L3, L22: A specific paragraph linking oxidative stress, inflammation and cytokines would be welcome

Done.

Figure 1 is really too synthetic. More details on the mechanisms for understanding the effects of polyphenols in the fight against oxidative stress would be welcome

P8L16-18, P9L6-9: the relationship between these structural features and the protective mechanism should be the subject of a Figure (with quercetin and its quinone and semiquinone forms) in place of Figure 2.

P10L40: As for flavonoids, a figure explaining the mechanisms by which stilbenes have a protective role against ROS is expected

A Table presenting the effects of quercitin in vivo and in vitro models, with doses and duration of use, is expected

A Table presenting the effects of resveratrol in vivo and in vitro models, with doses and duration of use, is expected

P13L7-13: as for the previous compounds, the mechanisms of action of curcumin should be the subject of a summary figure, the role of this compound (lipoxin inhibitor) on oxylipin formation should be better presented

The mechanisms and effects (alongside with the study model, dose and duration of treatment) of each polyphenol on each liver disease can now be found summarized in Table 1 (Quercetin), Table 2 (Resveratrol), and Table 3 (Curcumin).

P8L13, P9L16-37: the mechanisms by which ROS cause the production of pro-inflammatory cytokines should be detailed. The role of oxylipins in these processes should be summarised.

P12 L5-6: Here again, this mechanism and its relationship with the formation of oxylipins should be better detailed

The role of oxylipins was mentioned in page 15, lines 17-18 and lines 35-37.

Reviewer 3 Report

Title: 

Targeting oxidative stress with polyphenols to fight liver diseases

Authors: 

Machado I.F., Miranda R.G., Dorta D.J., Rolo A.P., Palmeira C.M.

Manuscript ID: antioxidants-2386336

Objective: 

This review provides an overview of oxidative stress and inflammation, including pathways right up to cancer. Regarding liver diseases like NAFLD,  (NAFL, NASH) and HCC, they focused on the beneficial effects of three polyphenols, i.e., quercetin, resveratrol and curcumin.

Points of criticism:

Page 1 – line 30:

… are mainly originated froFt3m… This needs to be amended.

Page 2 – first paragraph:

The authors describe the action of endogenous antioxidants, like catalase and peroxidases, concluding that their substrates were converted to water ignoring the whole reaction. This needs to be improved.

Page 3 – line 20:

…alterations in the intracellular AMP)to-adenosine triphosphate … This should be corrected.

Page 9 – line 35:

The authors refer to reference 146, i.e., Uylas et al. but indicated this reference erroneously with Uylia. This needs to be improved.

Page 10 – line 6:

… “thet”… – needs to be corrected.

Page 10 – line 21:

“Similarly, another in vitro and in vitro …” The authors presumably tried to indicate “in vitro and in vivo”. This should be amended.

Page 11 – lines 34:

“… The results were compared with existing studies on resveratrol supplementation with 500mg/day in patients with NAFLD but without T2DM, used for 12 weeks [173] and in another study for 3 months [167], …” 

This wording is misleading because 12 weeks and 3 months are identical. Therefore, this sentence should be rephrased.

Page 13 – line 15:

Ibrahin and colleagues should read “Ibrahim and colleagues” as indicated in Ref. 187.

General comment:

A graphic illustration of the diverse effects of polyphenols regarding different pathways would be worthwhile.  

Author Response

Reviewer 3

Objective:

This review provides an overview of oxidative stress and inflammation, including pathways right up to cancer. Regarding liver diseases like NAFLD,  (NAFL, NASH) and HCC, they focused on the beneficial effects of three polyphenols, i.e., quercetin, resveratrol and curcumin.

We thank the reviewer for the comments and suggestions. They were important to improve the quality of our manuscript.

All suggested corrections are highlighted in yellow in the manuscript.

Points of criticism:

Page 1 – line 30:

… are mainly originated froFt3m… This needs to be amended.

Corrected.

Page 2 – first paragraph:

The authors describe the action of endogenous antioxidants, like catalase and peroxidases, concluding that their substrates were converted to water ignoring the whole reaction. This needs to be improved.

Section 2 now discusses endogenous antioxidants in more detail (page 4, line 14).

Page 3 – line 20:

…alterations in the intracellular AMP)to-adenosine triphosphate … This should be corrected.

Page 9 – line 35:

The authors refer to reference 146, i.e., Uylas et al. but indicated this reference erroneously with Uylia. This needs to be improved.

Page 10 – line 6:

… “thet”… – needs to be corrected.

Page 10 – line 21:

“Similarly, another in vitro and in vitro …” The authors presumably tried to indicate “in vitro and in vivo”. This should be amended.

Page 11 – lines 34:

“… The results were compared with existing studies on resveratrol supplementation with 500mg/day in patients with NAFLD but without T2DM, used for 12 weeks [173] and in another study for 3 months [167], …”

This wording is misleading because 12 weeks and 3 months are identical. Therefore, this sentence should be rephrased.

Page 13 – line 15:

Ibrahin and colleagues should read “Ibrahim and colleagues” as indicated in Ref. 187.

We apologize for these typos. They are now corrected.

General comment:

A graphic illustration of the diverse effects of polyphenols regarding different pathways would be worthwhile. 

A table for each compound summarizing its mechanisms and effects on each liver disease was added to the manuscript.

Reviewer 4 Report

Thank you very much for submitting your manuscript to ANTIOXIDANTS.
For the treatment of liver disease, the authors analyzed various papers targeting oxidative stress with polyphenolic compounds and wrote a review paper. My comments are as follows:
1. As you know, several similar review papers appeared in PubMed. Therefore, it is necessary to specify why the authors selected three polyphenols and why these compounds were selected.
2. It is not very desirable to plot the structures of each of the three compounds, and more conclusive data is required. Rather, I recommend to present the results of the study by the authors and others, and to present future prospectives or research directions for which research direction/mission is desirable and how approaches can contribute to the treatment or prevention of liver diseases.
3. Based on the above comments, this review paper has not reached the acceptable limit for publishing in ANTIOXIDANTS.

.

Author Response

Reviewer 4

Thank you very much for submitting your manuscript to ANTIOXIDANTS.

For the treatment of liver disease, the authors analyzed various papers targeting oxidative stress with polyphenolic compounds and wrote a review paper. My comments are as follows:

We appreciate the comments and helpful suggestions made by the reviewer that were important to improve our work.

All suggested corrections are highlighted in yellow in the manuscript.

  1. As you know, several similar review papers appeared in PubMed. Therefore, it is necessary to specify why the authors selected three polyphenols and why these compounds were selected.

The justification for the selection of the three polyphenols was added to section 4.2 (page 9, line 15).

  1. It is not very desirable to plot the structures of each of the three compounds, and more conclusive data is required. Rather, I recommend to present the results of the study by the authors and others, and to present future prospectives or research directions for which research direction/mission is desirable and how approaches can contribute to the treatment or prevention of liver diseases.

We thank the reviewer for the suggestions. The structures of the compounds were removed from the manuscript. The section Concluding remarks was complemented with future perspectives.

Round 2

Reviewer 2 Report

Thank you for this additional information which, I believe, significantly improves the attractiveness of your very interesting review of the literature.

Reviewer 4 Report

Thank you for your revision. Answers to major comments are considered sufficeint. There are still lots of misspellings. 

Thank you for your revision. Answers to major comments are considered sufficeint. There are still lots of misspellings.